# Optimization of Extraction Process and Activity of Angiotensin-Converting Enzyme (ACE) Inhibitory Peptide from Walnut Meal

**DOI:** 10.3390/foods13071067

**Published:** 2024-03-30

**Authors:** Meng Meng, Ziyi She, Yinyin Feng, Junhan Zhang, Ran Han, Yanlong Qi, Lina Sun, Huiqing Sun

**Affiliations:** 1State Key Laboratory of Food Nutrition and Safety, College of Food Science and Engineering, Tianjin University of Science and Technology, No. 29, 13th Avenue, Tianjin Economy Technological Development Area, Tianjin 300457, China; mengmeng@tust.edu.cn (M.M.); 15931795027@163.com (Z.S.);; 2Key Laboratory of Food Nutrition and Safety, Ministry of Education, Tianjin University of Science and Technology, No. 29, 13th Avenue, Tianjin Economy Technological Development Area, Tianjin 300457, China; 3College of Food Science and Engineering, Tianjin University of Science and Technology, No. 29, 13th Avenue, Tianjin Economy Technological Development Area, Tianjin 300457, China; 4Research Institute of Farm Products Storage and Processing, Xinjiang Academy of Agricultural Sciences, No. 403 Nanchang Road, Urumqi 830091, China; qiyanlong9@126.com; 5Institute of Agricultural Mechanization, Xinjiang Academy of Agricultural Sciences, No. 291 South Nanchang Road, Urumqi 830091, China

**Keywords:** walnut meal, enzymatic hydrolysis, ACE inhibition rate, anti-oxidation, stability

## Abstract

In order to further realize the resource reuse of walnut meal after oil extraction, walnut meal was used as raw material to prepare polypeptide, and its angiotensin-converting enzyme (ACE) inhibitory activity was investigated. The ACE inhibitory peptides were prepared from walnut meal protein by alkaline solution and acid precipitation. The hydrolysis degree and ACE inhibition rate were used as indexes to optimize the preparation process by single-factor experiment and response surface method. The components with the highest ACE activity were screened by ultrafiltration, and their antioxidant activities were evaluated in vitro. The effect of gastrointestinal digestion on the stability of walnut peptide was analyzed by measuring molecular weight and ACE inhibition rate. The results showed that the optimal extraction conditions were pH 9.10, hydrolysis temperature 54.50 °C, and hydrolysis time 136 min. The ACE inhibition rate of walnut meal hydrolysate (WMH) prepared under these conditions was 63.93% ± 0.43%. Under the above conditions, the fraction less than 3 kDa showed the highest ACE inhibitory activity among the ACE inhibitory peptides separated by ultrafiltration. The IC_50_ value of scavenging ·OH free radical was 1.156 mg/mL, the IC_50_ value of scavenging DPPH free radical was 0.25 mg/mL, and the IC_50_ value of scavenging O_2_^−^ was 3.026 mg/mL, showing a strong total reducing ability. After simulated gastrointestinal digestion in vitro, the ACE inhibitory rate of walnut peptide decreased significantly, but it still maintained over 90% ACE inhibitory activity. This study provides a reference for the application of low-molecular-weight walnut peptide as a potential antioxidant and ACE inhibitor.

## 1. Introduction

Walnut has a long planting history in China and a wide distribution range. Xinjiang walnut planting area belongs to the northwest of the four planting areas in our country. The planting area and yield of walnut have certain scale advantages compared with other provinces in China. Walnut is rich in high-quality protein, oil, cellulose, vitamins and other essential nutrients for the human body. Its use and medicinal value make walnut a food resource with great development prospects [1]. At present, the research is mainly focused on walnut oil. The protein content of walnut meal after oil extraction is more than 40%, but it has not been fully used, which not only wastes resources but also causes pollution. To enhance the depth of processing efficiency and overall utilization rate of walnut meal, further processing of peptides derived from readily accessible walnut meal was undertaken. Current research on the active ingredients in walnuts mainly focuses on the extraction of polyphenols, flavonoids, proteins, and walnut oil, as well as their antioxidant, antibacterial, anti-inflammatory, anti-cancer, and memory improvement effects [2]. Current research efforts center on identifying natural bioactive compounds that can effectively reduce blood pressure, with ACE inhibitory peptides gaining significant attention. In this context, researchers such as Dr. Kunlun Liu’s team have successfully isolated and characterized four distinct peptides from maize embryo proteins, demonstrating ACE inhibitory properties. They delved deeper into understanding the unique inhibitory mechanisms of these peptides through kinetic studies and molecular docking analysis [3]. Duan et al. identified three novel ACE-inhibiting peptides by screening rapeseed protein sequences in a database using bioinformatics methods and synthesized and verified their biological activities [4].

Angiotensin-converting enzyme (ACE), a zinc dipeptide carboxypeptidase, is a key target in the pathogenesis of hypertension [5]. ACE plays a critical role in the human blood pressure regulatory system by catalyzing the cleavage of two terminal amino acids from angiotensin I to form angiotensin II, thereby inducing vasoconstriction and contributing to elevated blood pressure levels [6]. It can also cause blood pressure to rise through degradation of bradykinin, which deactivates it [7]. ACE inhibitors inhibit the activity of ACE to block the pathway that increases blood pressure to achieve the purpose of lowering blood pressure. At present, numerous synthetic antihypertensive medications, including captopril, lisinopril, and eplerenone, are employed in the clinical management of hypertension. However, effective long-term use of these medications can cause a variety of adverse side effects like coughing, rashes, diarrhea, high potassium levels, kidney issues, and taste alterations, which can gradually erode patient well-being [8]. ACE inhibitory peptides extracted from natural plants and animals not only have good antihypertensive efficacy, but also are non-toxic and have no side effects. Therefore, foodborne ACE inhibitory peptides have good application prospects.

The enzymatic hydrolysis product with high ACE inhibition activity was obtained from the walnut cake after oil extraction of Xinjiang walnut by alkaline protease enzymatic hydrolysis. Taking the enzymatic hydrolysis product as the research object, the ACE inhibition activity was used as the evaluation index, and the extraction conditions were optimized by the Box–Behnken test. Under optimal conditions, the peptide fraction was further separated by ultrafiltration to explore its antioxidant activity. In vitro digestion simulation was performed to study the gastrointestinal stability. This study can provide a reference for further development and utilization of walnut protein and other plant proteins.

## 2. Materials and Methods

### 2.1. Materials

The walnut cultivar was Xinjiang Wen185, which was produced in Wensu County, Aksu, Xinjiang.

Alkaline protease (20,000 U/g), angiotensin converting enzyme, Hippuryl-His-Leu-OH (HHL), pepsin, trypsin—Beijing Solaibao Technology Co., LTD., Beijing, China; N-hexane, sodium hydroxide, hydrochloric acid, boric acid, ethyl acetate, anhydrous ethanol, ferric chloride, potassium ferricyanide (all analytical pure)—Shanghai McLean Biochemical Technology Co., LTD., Shanghai, China; phenol, trichloroacetic acid (all analytically pure) —Shanghai Jizhi Biochemical Technology Co., LTD., Shanghai, China.

### 2.2. Preparation of Defatted Walnut Meal

Once the walnut meal has undergone crushing and subsequent sieving through an 80-mesh screen, it is mixed with n-hexane in a solid-to-liquid ratio of 1:5 (g/mL). The blend is then allowed to sit in a water bath maintained at a temperature of 45 °C for a duration of 1 h. Then, it is drained and filtered. The filter cake is collected and extracted again until the filtrate is colorless and transparent, and it is completely dried in an oven at 60 °C.

### 2.3. Preparation of Walnut Protein

Deionized water was incorporated into the nonfat walnut powder following a material-to-liquid ratio of 1:20 and thoroughly blended. Subsequently, the pH level of the mixture was adjusted up to 11, allowing for complete dissolution of the protein in the walnut powder under magnetic stirring at 53 °C for an interval of 1.5 h. After cooling, the mixture is centrifuged at 5000× *g* of centrifugal force for 15 min, and then the supernatant is carefully poured out.

### 2.4. Optimization of Preparation Process of Walnut Peptide by Enzymatic Hydrolysis

#### 2.4.1. Single-Factor Test

The substrate concentration of 2% walnut protein was added into ultrapure water, and the substrate of alkaline protease 8000 U/g was added; the enzymatic hydrolysis temperature was 55 °C. During the enzymatic hydrolysis, the pH of the solution was kept constant at 9. Upon completion of a 2-h enzymatic hydrolysis period, the enzyme activity was terminated by boiling the mixture for 10 min. After the cooling phase, the supernatant was centrifuged at a centrifugal force of 7000× *g* for 15 min. The single-factor test factors and levels were presented in the Table 1.

#### 2.4.2. Response Surface Optimization Method

According to the initial findings from the one-factor-at-a-time tests, three significant variables were identified and selected for advanced optimization using response surface methodology. These factors included pH (Factor A), enzymatic temperature (Factor B), and enzymatic time (Factor C), utilizing the ACE inhibitory efficiency as the response parameter to gauge the outcomes. To systematically optimize these factors, a Box–Behnken experimental design was implemented. Table 2 presents the precise parameter settings and levels for each of these influencing factors in the designed experiment.

### 2.5. Degree of Hydrolysis

The degree of hydrolysis (DH) was determined employing the pH-stat method, representing the proportion of peptide bonds that have been cleaved during the enzymatic hydrolysis process [9]. In the experiment, the pH of the enzymolysis system was kept at a stable level by adding NaOH solution continuously. DH is determined by calculating the amount of NaOH volume that needs to be added throughout the process, and the NaOH volume is proportional to the degree of cleavage of the peptide bond.
$$DH(\%)=\frac{B\times N_{b}}{Mp\times \alpha \times h_{tot}}\times 100$$
where: *B* is the volume of consumed NaOH standard solution, mL; *N_b_* is the concentration of NaOH standard solution mol/L; *α* is the degree of dissociation of the amino group, where the value is 1; *Mp* is the total amount of protein in the substrate, g; *h_tot_* is the millimolar number of substrate protein bonds per gram, mmol/g, and this value is taken to be 8.0 mmol/g.

### 2.6. Evaluation of ACE Inhibitory Activity

ACE (50 mU/mL), along with the substrate HHL (8 mmol/mL), were both solubilized in a borate buffer solution with a pH of 8.3. The components were then successively introduced into separate 1.5 mL centrifuge tubes as outlined in Table 3 and subsequently incubated in a water bath according to the prescribed conditions.

After the reaction is complete, 0.8 mL of ethyl acetate is introduced into the blend, followed by 20 s of agitation, and then precipitated for liquid separation. Then, carefully remove the 0.6 mL ethyl acetate layer and dry it in an oven heated to 100 °C. After cooling, 0.3 mL of deionized water was added to the dry tube, vigorously rotated for 30 s, and finally, the absorbance was measured at 228 nm wavelength.
$$ACE\;inhibitory\;activity(\%)=\frac{A_{b}-A_{a}}{A_{b}-A_{c}}\times 100$$
where: *A_a_* is the absorbance of the experimental group; *A_b_* is the absorbance of the control group; and *A_c_* is the absorbance of the blank group.

### 2.7. Ultrafiltration Separation

The products were separated using centrifugal ultrafiltration tubes with molecular weight cutoffs of 10 kDa and 3 kDa. Initially, the membranes of these tubes were immersed in ultrapure water on ice for a period of 10 min. Subsequently, the water was discarded, and 10 mL of the hydrolyzed solution was loaded into each tube. The tubes were then centrifuged at a force of 5000× *g* under a temperature of 4 °C for a duration of 20 min.

### 2.8. Determination of Antioxidant Capacity

#### 2.8.1. ·OH Free Radical Scavenging Ability

The ·OH radical scavenging capacity was assessed using the Fenton reaction system method. To carry out the process, it was necessary to introduce 0.1 mL of the peptide solution into a centrifuge tube with a capacity of 1.5 mL. Following this, 0.1 mL of a ferric sulfate solution with a concentration of 6 mmol/L and an equal amount of hydrogen peroxide solution were put into the tube. The last step involved topping off the mixture with 1 mL of a salicylic acid-ethanol solution with a concentration of 6 mmol/L. The solution was subjected to incubation in a water bath regulated at 37 °C for a period of one hour. Following this, we used a wavelength of 510 nm to determine the absorbance value *A*_1_. To determine the absorbance values *A*_0_ and *A*_2_ we substituted the sample solution and hydrogen peroxide solution with distilled water, respectively. The experiment was conducted in three parallel trials, using Vc as our positive control.

#### 2.8.2. DPPH Free Radical Scavenging Ability

To evaluate the capability of scavenging DPPH free radicals, the peptide solution (0.2 mL) was combined with DPPh-ethanol solution (0.2 mL, 0.2 mmol/L) and vortexed for thorough mixing. The mixture was then left to stand in the dark for 30 min, and the absorbance was subsequently measured at a wavelength of 517 nm, which was recorded as *A*_1_. The absorbance values were recorded as *A*_0_ and *A*_2_ by anhydrous ethanol instead of polypeptide solution and DPPH-ethanol solution, respectively. With Vc as the positive control, the experiment was repeated in parallel 3 times.

#### 2.8.3. O_2_^−^ Scavenging Ability

The antioxidant capacity was evaluated by pyrogallol autoxidation method. Tris-HCl buffer 0.3 mL (pH 8.2, concentration 0.5 M) was prepared for the test. Then, 0.1 mL peptide solution was added, and after 20 min of water bath at 25 °C, 0.04 mL preheated pyrocatechol solution (25 mmol/L) was added. The solution underwent a rotation to ensure its accurate reaction for a duration of four minutes. Upon cessation of the reaction, a concentrated hydrochloric acid volume of 0.05 mL was promptly added. Following this, the resultant sample underwent centrifugation at 7000× *g* at a temperature of 4 °C for a period of three minutes. The absorbance was then measured at 325 nm and denoted as *A*_1_. The absorbance values obtained by using distilled water instead of sample solution and pyrotriol solution were denoted as *A*_0_ and *A*_2_, respectively. The whole procedure was repeated three times with Vc as a positive control.
$$free\;radical\;rate\;of\;clearance/\%=\left(1-\frac{A_{1}-A_{2}}{A_{0}}\right)\times 100$$

#### 2.8.4. Total Reducing Power

The Prussian Blue Assay method was utilized to evaluate the overall reducing capability. During this procedure, a 0.1 mL quantity of the peptide solution was dispensed into a 1.5 mL centrifuge tube, followed by the addition of 0.25 mL of phosphate buffer (adjusted to pH 6.6 and 0.2 Molar concentration). Next, 0.25 mL of a 1% potassium ferricyanide solution was mixed in. Subsequently, 0.25 mL of a 10% trichloroacetic acid solution was introduced. After thorough vortex mixing, the reading was taken at a wavelength of 700 nm. Parallel experiments were performed three times with Vc as a positive control. A higher OD value indicates a stronger total reducing power of the test sample.

### 2.9. Simulating Gastrointestinal Digestion In Vitro

The method of Zhang Li for in vitro simulation of gastrointestinal digestion was slightly modified [10]. The fractions smaller than 3 kDa were initially separated and then mixed with the simulated gastric fluid at a ratio of 1:20 (mass/volume). This simulated fluid comprised a combination of equal volumes of 0.02 mol/L NaCl, 0.02 mol/L KCl, and 0.02 mol/L NaHCO_3_. The pH level of the mixture was modified to 1.5 through the inclusion of 0.5 mol/L HCl, which facilitated the dissolution of 4% pepsin. Prior to the introduction of the peptide solution, the solution was subjected to a pre-incubation period of 5 min at a temperature of 37 °C. The hydrolysis process was carried out at 37 °C for a duration of 2.5 h, following which enzyme activity was terminated by heating the samples in boiling water at 100 °C for 10 min.

The pH of the remaining solution was re-set to 7.5 using a 1 mol/L NaOH solution. After that, 4% trypsin was added, and the solution was subjected to another round of digestion for 2.5 h at a temperature of 37 °C. The activity of the enzyme was then stopped through heat inactivation by boiling the sample for 10 min after collection. After cooling down, centrifuge the mixture at 7000× *g* for 15 min at a temperature of 4 °C. A portion of the supernatant was freeze dried for measuring ACE inhibitory activity, while the remainder was stored in a −20 °C refrigerator for future use.

### 2.10. Determination of the Relative Molecular Mass of Peptides

The molecular mass distribution was analyzed by HPLC [11]. The enzyme solution of walnut meal was prepared into a 2 mg/mL solution and passed through a 0.45 μm filter membrane. The TSK gel 2000 SWXL chromatographic column with dimensions of 7.8 mm × 30 cm was utilized, where the mobile phase comprised a ratio of acetonitrile, water, and trifluoroacetic acid in volumes of 10:40:0.05, respectively. The detection wavelength was 214 nm, the temperature recorded by the column stood at 30 °C, while the flow rate was clocked at 0.5 mL/min. Moreover, the injection volume was noted to be 10 μL. LC solution Version 1.22 SP1 was used for automatic data processing and calculation.

To prepare a molecular weight standard curve, five different peptide standards with varying molecular weights were employed: Cytochrome C (Mw = 12,200 Da), Aprotinin (Mw = 6511.44 Da), Bacitracin (Mw = 1421.69 Da), Gly-Gly-Tyr-Arg (Mw = 451.48 Da), and Ala-Ala-Ala (Mw = 189.17 Da). The curve was constructed to calculate the distribution of peptide molecular weights in the sample.

## 3. Results

### 3.1. Single-Factor Test of Enzymatic Hydrolysis Process

#### 3.1.1. Enzymatic Effect

As shown in Figure 1, with the increment of enzyme dosage, there was a corresponding increase in the binding of the enzyme to the substrate, consequently leading to a heightened rate of enzymatic reaction. When the enzyme concentration was 6000 U/g, the DH value reached 16.74%, the ACE inhibition rate attained a considerable 63.85% mark at an enzyme concentration of 6000 U/g., the binding between enzyme and substrate was approximately saturated, and the reaction rate tended to be flat. Therefore, 6000 U/g was selected as the condition of enzyme addition in subsequent experiments.

#### 3.1.2. Effect of Temperature

As shown in Figure 2, the degree of hydrolysis first increased and then decreased during the temperature rise. The highest inhibition rates of DH and ACE were 16.73% and 63.65%, respectively, when the temperature was 55 °C. However, t as the temperature for enzymatic hydrolysis kept escalating, and a decrease in the efficiency of the process was observed; it is possible that the deleterious effects of high temperatures on the enzyme’s molecular stability and the ability of its binding sites to attach to the substrate may be the root cause of the issue, thus causing a drop in the degree of hydrolysis. Consequently, 55 °C was determined as the optimal enzymatic hydrolysis temperature for the subsequent single-factor experiments.

#### 3.1.3. Effect of pH

The inhibition activity of ACE demonstrated an upward trend initially, followed by a decline, with the increase in pH level (Figure 3). The peak of ACE inhibition rate was recorded at a pH value of 9. DH increased continuously from pH 8.5 to 10 and then decreased. Raising the pH level can impede the optimal performance of the enzyme by interrupting its association with the substrate, which consequently lessens the extent of hydrolysis. Nevertheless, the highest degree of ACE inhibition was observed at pH 9, suggesting that ACE inhibition was not directly related to the degree of hydrolysis. Therefore, in order to protect the potency of the peptide from any harmful effects of excess ·OH, and to guarantee a robust hydrolysis while not hindering the peptide’s activity, it is vital to sustain a moderate pH level throughout the enzymatic process. Finally, the optimal pH of enzymatic hydrolysis in this experiment was determined to be 9 as the condition for subsequent single-factor experiments.

#### 3.1.4. Effect of Time

The results in Figure 4 showed that the DH and ACE inhibition rates increased with the prolongation of the enzymatic hydrolysis time. Within the initial 2 h, the reaction progressed swiftly, with the enzyme rapidly bonding to the substrate during the early stages of enzymatic hydrolysis. However, beyond the 2-h mark, the increase in both DH and the ACE inhibition rate began to slow down. At the 2-h mark of enzymatic hydrolysis, the ACE inhibitory activity had reached 63.17%. By the 3-h point, it further increased marginally to 63.88%. The enzymatic hydrolysis time of 2 h was selected as the optimal condition in terms of saving time and cost.

### 3.2. Response Surface Optimization of Enzymatic Hydrolysis Process

#### 3.2.1. Response Surface Experiments—Design and Results

Based on the findings from the single-factor tests where pH, hydrolysis temperature, and hydrolysis time were treated as independent variables, to enhance the preparation of ACE inhibitory peptides via the alkaline protease enzymatic hydrolysis of walnut protein, a response surface methodology experiment was undertaken, with the ACE inhibition rate serving as the dependent response variable. The primary objective was to refine and optimize the process parameters, resulting in an ideal outcome. The detailed experimental design and outcomes are presented in Table 4.

#### 3.2.2. Response Surface Regression Equation

The regression equation was fitted using Design Expert 12, wherein the three independent variables of pH, enzymatic hydrolysis temperature, and enzymatic hydrolysis time were represented by A, B, and C, respectively. The quadratic multinomial regression model was obtained through multiple fitting regression of the ACE inhibition rate: Y = 63.67 + 0.915A − 0.7175B + 1.64C + 0.865AB + 0.61AC + 0.295BC − 2.23A^2^ − 1.81B^2^ − 1.66C^2^.

Table 5 illustrates the outcomes of the regression analysis of the rate model for ACE inhibition and the correlation regression coefficients. If the *p*-value of the regression model is less than 0.01, it is considered to be highly statistically significant. Moreover, the lack-of-fit term’s *p*-value stands at 0.97, exceeding the 0.05 benchmark, hence indicating that the model provides a good fit for the data and can accurately predict the corresponding regression value from the equation with consistent reliability. The *p*-values of A (pH), B (hydrolysis temperature), and C (hydrolysis time) were all less than 0.01, indicating that they had significant effects on ACE inhibition rate. The joint impact of pH and enzymatic hydrolysis temperature on ACE inhibition rate was found to be considerable, as the interaction term AB’s *p*-value was less than 0.01. Furthermore, the *p*-value of AC term (interaction between pH and enzymatic hydrolysis time) was less than 0.05, indicating that their interaction also significantly affected ACE inhibition rate. In contrast, B (enzymolysis temperature) and C (enzymolysis time) had no significant effect on ACE inhibition rate.

At the same time, the regression coefficient R^2^ = 0.9844 was basically consistent with the adjusted regression coefficient R^2^ = 0.9644; the test results demonstrate a strong concurrence between the model fitted and the actual data, which could be used to explain 96.44% of the data. The equation had high credibility, the signal-to-noise ratio was small (C.V.% = 0.73), and the data-fitting degree was good with small differences. The F-value reflects the influence relationship of A single factor on ACE inhibition rate; the causal relationship between the factors can be ranked in order of enzyme hydrolysis time (C) having the greatest impact, followed by pH (A), and then enzyme hydrolysis temperature (B).

#### 3.2.3. The Results of Interaction

As both the pH value and enzymatic temperature rise concurrently, the ACE inhibition rate of the enzymatic solution initially shows an increase, followed by a subsequent decline, as depicted in Figure 5, affirmatively demonstrating that the interaction between pH and enzymatic temperature exerts a highly significant impact on the ACE inhibition rate. Referring to Figure 6, the contour plot depicting the interaction between pH and digestion time reveals an elliptical shape, with a steep response surface gradient. This indicates that both pH and its interaction with the enzyme solution temperature have substantial effects on the ACE inhibition rate, which aligns with the findings from the variance analysis.

#### 3.2.4. Verification of the Optimal Enzymatic Hydrolysis Process

Response surface analysis showed that the optimal process conditions for ACE inhibition of peptidase hydrolysis were pH 9.13, hydrolysis temperature 54.54 °C, and hydrolysis time 136.04 min, and the highest ACE inhibition rate was 64.26% under these conditions. Considering the actual situation, the extraction process parameters were adjusted. The optimal enzymatic hydrolysis process conditions were pH 9.10, enzymatic hydrolysis temperature 54.50 °C, and enzymatic hydrolysis time 136.00 min. The ACE inhibition rate of walnut protease hydrolysate obtained was 63.93% ± 0.43%, which was within 1% of the theoretical prediction value.

### 3.3. Ultrafiltration Separation

Ultrafiltration is a membrane separation technique that separates solutes based on their molecular weight. The enzymatic hydrolysates were separated with ultrafiltration centrifuge tubes of different molecular weight sizes to obtain the preliminarily isolated peptide fractions. The ultrafiltration results showed (Table 6) that the yield of WRH-ΙΙ (3000~10,000 Da) was higher at 39.82%. However, it is noteworthy that the sample’s ACE inhibition rate is markedly inferior in comparison to that of WMH-I, and its IC_50_ for ACE inhibition is 0.299. When the concentration is 1 mg/mL, the ACE inhibition rate obtained by the experiment is only 63.93%, which is significantly lower than the data obtained by Liu et al. [12]. The literature proved that the inhibition rate of ACE reached as high as 82.4% in components with molecular weight less than 3 kDa. Given this significant difference, we decided to proceed with WMH-I for our subsequent investigations.

### 3.4. In Vitro Antioxidant Activity

#### 3.4.1. ·OH Scavenging Ability

As shown in Figure 7, the capacity to remove ·OH gradually increased as the concentration of WMH-Ι rose from 1 mg/mL to 6 mg/mL, resulting in a sharp increase in the clearance rate when the concentration was between 1 and 3 mg/mL. At a concentration below 6 mg/mL, the ·OH clearance rate for Vc was 96.6%, while that for WMH-Ι was slightly lower at 82.6%. This variation suggests that WMH-Ι’s performance, while lower than that of Vc, boasts an impressive capacity for ·OH elimination. The IC_50_ value of WRH-Ι for ·OH clearance stood at 1.156 mg/mL. 

#### 3.4.2. DPPH Scavenging Ability

In Figure 8, it is evident that between the concentrations of 0.1 mg/mL and 0.6 mg/mL, a noticeable trend can be discerned. Both Vc and WMH-Ι demonstrate an escalating trend in their scavenging capabilities against DPPH free radicals as the sample concentration rises. Notably, between concentrations of 0.1 mg/mL and 0.4 mg/mL, WMH-Ι displays a marked increase in its ability to scavenge DPPH rapidly. At the concentration of 0.6 mg/mL, Vc displayed an exceptional DPPH clearance rate of 94.2%, while WMH-Ι displayed a clearance rate of 81.1% for the identical radicals, albeit being slightly less efficient compared to Vc in terms of its DPPH clearance ability. Despite this difference, these results still highlight a robust scavenging ability of WMH-Ι against DPPH radicals. The determined IC_50_ measurement for the DPPH radical scavenging capability of WMH-1 revealed a value of 0.25 mg/mL.

#### 3.4.3. O_2_^−^ Scavenging Ability

O_2_^−^ is a weak free radical that is produced by mitochondrial electron transport. The aforementioned is also regarded as an antecedent to the production of reactive oxygen species (ROS), which is accountable for the manifestation of a plethora of ailments [13]. As shown in Figure 9, the content of WMH-Ι demonstrates a fluctuation, ranging from 0.2 mg/mL to 1.2 mg/mL; there is a noticeable enhancement in its superoxide anion scavenging rate in correspondence with the increase in concentration. However, the rate of increase follows a relatively modest slope, showing a gradual and cumulative improvement that adheres to a dose-dependent pattern. Regarding the scavenging activity against O_2_^−^, the calculated IC_50_ value for WMH-Ι is 3.026 mg/mL, which reflects the concentration needed to achieve 50% scavenging efficiency for O_2_^−^.

#### 3.4.4. Total Reducing Power

As shown in Figure 10, as the concentration increased from 1 mg/mL to 6 mg/mL, there was a corresponding increase in the overall reducing capacities of both Vc and WMH-Ι. Nonetheless, it is noteworthy that throughout this concentration range, WMH-Ι consistently demonstrated a significantly lesser reducing power compared to that of Vc. The results showed that WRH-Ι has a certain total reducing capacity.

### 3.5. Changes in ACE Inhibition Rate during Digestion

The results of changes in ACE inhibition rate during digestion were illustrated in the Figure 11. The residence time of food in the stomach is generally 2–2.5 h, pepsin is present in gastric juice, and the pH value ranges from 1 to 2. After evaluating the rate of ACE inhibition at a concentration of 2 mg/mL during the process of digestion, it was observed that the inhibition rate experienced a noteworthy decrease (*p* < 0.05) following 2.5 h of simulated gastric digestion. However, even post-decrease, the ACE inhibition rate remained appreciable at 60.41% ± 1.27%. Further, following 2.5 h of intestinal digestion, there was another significant decline (*p* < 0.05) in the ACE inhibition rate, yet it still managed to reach 63.66% ± 0.74%. The inhibition rate of ACE was also significantly decreased when simulated gastrointestinal digestion was performed at the same time (*p* < 0.05), but the inhibitory activity of ACE was higher than that of single treatment. After digestion, the inhibitory activity of the peptide ACE decreased, which was similar to the change pattern studied by Zhenqiu Xu et al. [14]. However, in spite of that, Osama Magouz and his colleagues reported divergent findings, in which they found a significant increase in the ACE inhibitory activity of the fermented buttermilk-derived peptides post-gastrointestinal digestion [15]. During the gastrointestinal digestion process, enzymatic processes instigated by molecules like pepsin and trypsin are responsible for breaking down peptides into smaller peptide fragments or singular amino acids. This enzymatic breakdown can lead to the disruption of certain structural elements that possess ACE inhibitory properties, thereby contributing to a decrease in overall ACE inhibitory activity [16]. The short peptides produced in different digestion processes are inconsistent, which inevitably leads to different degrees of changes in their inhibition rates.

### 3.6. Changes in Molecular Mass Distribution during Digestion

The distribution of molecular masses smaller than 3 kDa was analyzed by HPLC for different digestion processes. As shown in Figure 12, the percentage of peptides with a molecular weight below 1000 Da in the digested hydrolysate underwent a notable augmentation, mirroring the findings of Zhengli Lin’s investigation [11]. The extent of reduction in the share of peptides exhibiting a molecular weight of over 2000 Daltons varied. During digestion, trypsin acts on the peptide bond linked by lysine and arginine, which further hydrolyze the peptide of 1000 to 2000 Da into smaller fragments. Wen Wenjun showed that the molecular weight of wheat was in the range of 70 to 3000 Da after gastrointestinal digestion, and it was a peptide with effective ACE inhibitory activity [17].

## 4. Discussion

In recent years, there has been a growing interest in ACE-inhibiting peptides sourced from natural origins, including those extracted from both animals and plants. However, they are often ignored due to their low bioavailability in vivo. In the present study, walnut protein hydrolysates treated with alkaline protease showed higher ACE inhibitory activity after optimized extraction conditions. We have verified that the protein hydrolysate’s ACE inhibitory function is unequivocally associated with the molecular weight of its constituents; hence, we proceeded to fractionate the walnut hydrolysate based on the varying sizes of its molecular weights.

The smallest molecular weight fraction (<3 kDa) of hydrolyzed walnut protein exhibited the greatest ACE inhibitory activity. Peptides with low molecular weight are more likely to bind to target molecules than peptides with high molecular weight and therefore exhibit higher ACE inhibitory activity [18]. Consistent with the earlier scientific literature, our investigation confirms that peptides of lower molecular weights generally display a higher potency in terms of inhibiting ACE activity [10]. In addition, the process of ultrafiltration aids in preserving hydrophobic amino acids at the terminal position of the peptide, promoting inhibition [19].

Antioxidants function to impede or mitigate the oxidative harm inflicted upon cells and tissues within the body. These free radicals, characterized by their heightened reactivity, can inflict damage on cellular components such as proteins and DNA, ultimately causing oxidative stress and injury [20]. O_2_^−^ serves as the initial reactive oxygen species engendered in the human body, setting off the generation of a host of other ROS. ROS play integral roles in nearly all biological and disease-related processes occurring within living organisms, with O_2_^−^ often recognized as a key responsive element in these mechanisms [21]. The superoxide anion, being highly reactive, readily transforms into various alternative forms of ROS, including peroxy nitrite, ·OH, and hydrogen peroxide, among others [22]. 

The state of oxidative stress results from the imbalance between oxidation and the body’s antioxidant capacity, and ROS levels in the system are elevated [23]. Excess ROS can rapidly react with nitric oxide (NO) to produce peroxynitrites that are highly toxic to biofilms and intracellular structures [24]. This results in a loss of the effectiveness of NO, which disrupts the NO-mediated vasodilation. It further aggravates endothelial dysfunction and ultimately promotes the occurrence and development of hypertension [25]. Therefore, the antioxidant effect of ACE inhibitors contributes to the prevention and improvement of hypertension.

Bioactive compounds present in food play a vital role in the absorption process of the gastrointestinal tract and can only showcase their physiological effects once they are assimilated into the bloodstream [26]. Despite evidence showing that numerous food-originating peptides can indeed be fully absorbed, it remains true that a significant number of proteins continue to be susceptible to degradation by proteases located on the surface of intestinal microvilli [27]. It has been reported that dipeptides and tripeptides can be absorbed by the body entirely through intestinal endothelial cells [28]. Therefore, the assessment of the durability of ACE-inhibiting peptides during the process of digestion establishes a theoretical foundation for utilizing them as functional elements in food products. This approach aims to enhance the efficient absorption and availability of these active peptides in the human system.

## 5. Conclusions

In this study, the enzymatic preparation process of walnut peptidase was optimized by single-factor test and response surface test, and the optimal process conditions were pH 9.10, enzymatic temperature 54.50 °C, and enzymatic time 136 min. Under these conditions, the ACE inhibition rate of the enzymatic hydrolysate reached 63.93% ± 0.43%, and the IC_50_ value of ACE activity of fractions less than 3 kDa obtained after ultrafiltration was 0.299. The IC_50_ value of ·OH, DPPH radical, and O_2_^−^ was 1.156 mg/mL, 0.25 mg/mL, and 3.026 mg/mL, respectively, and it has strong antioxidant capacity and total reducing power in vitro. ACE inhibition retention rate was more than 90% after simulated gastrointestinal digestion in vitro, which provided reference for further development of ACE-inhibiting peptides in walnut.

## Figures and Tables

**Figure 1 foods-13-01067-f001:**
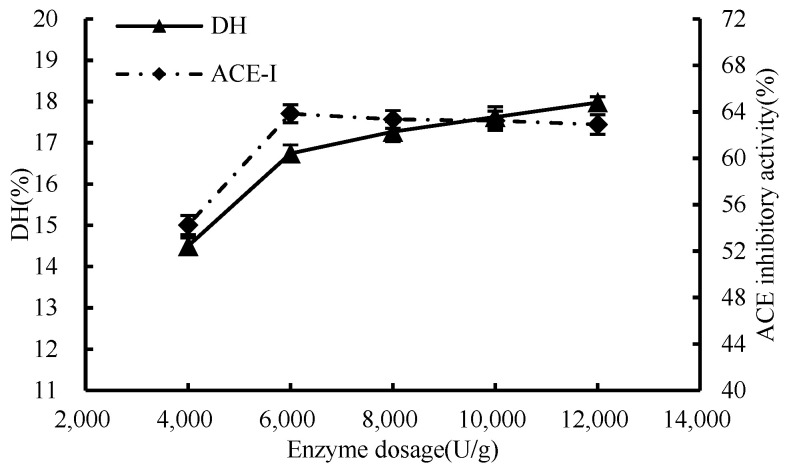
Effect of enzyme dosage on hydrolysis degree and ACE inhibition rate.

**Figure 2 foods-13-01067-f002:**
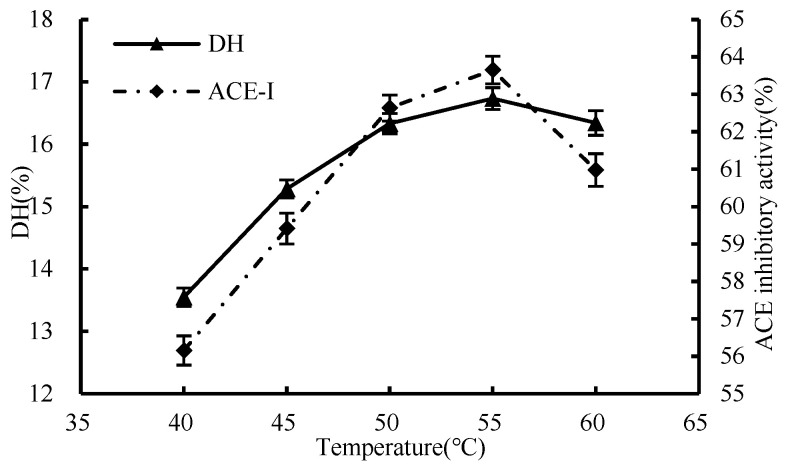
Influence of enzymatic hydrolysis temperature on hydrolysis degree and ACE inhibition rate.

**Figure 3 foods-13-01067-f003:**
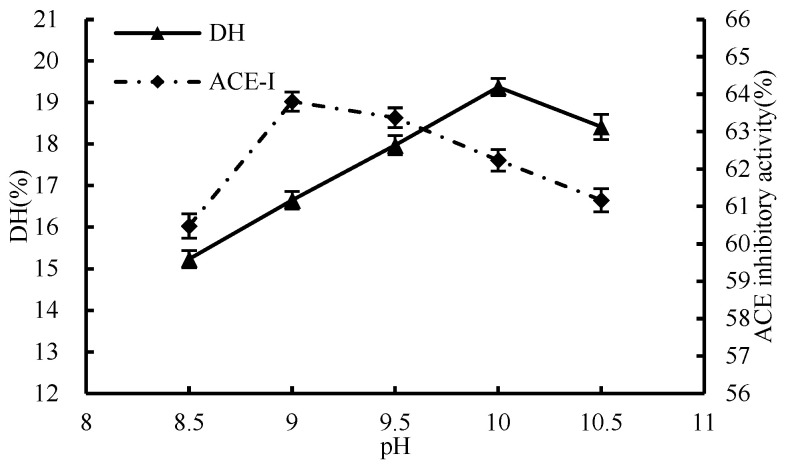
Effect of pH on hydrolysis degree and ACE inhibition rate.

**Figure 4 foods-13-01067-f004:**
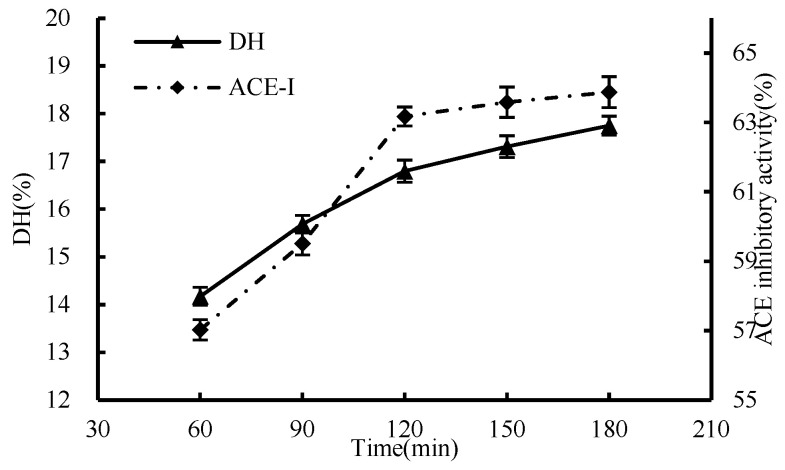
Effect of enzymatic hydrolysis time on hydrolysis degree and ACE inhibition rate.

**Figure 5 foods-13-01067-f005:**
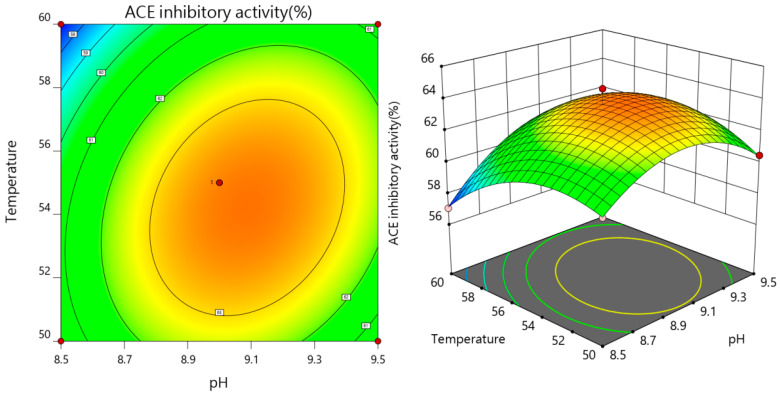
Effect of pH and temperature of enzymatic hydrolysis on the rate of ACE inhibition.

**Figure 6 foods-13-01067-f006:**
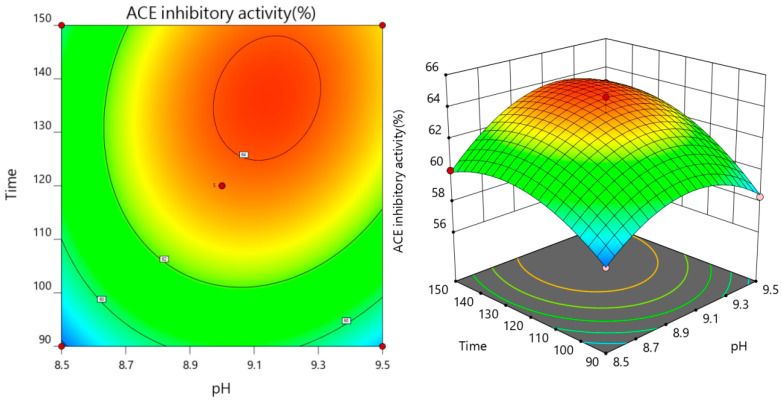
Interaction of pH and enzymatic hydrolysis time on ACE inhibition rate.

**Figure 7 foods-13-01067-f007:**
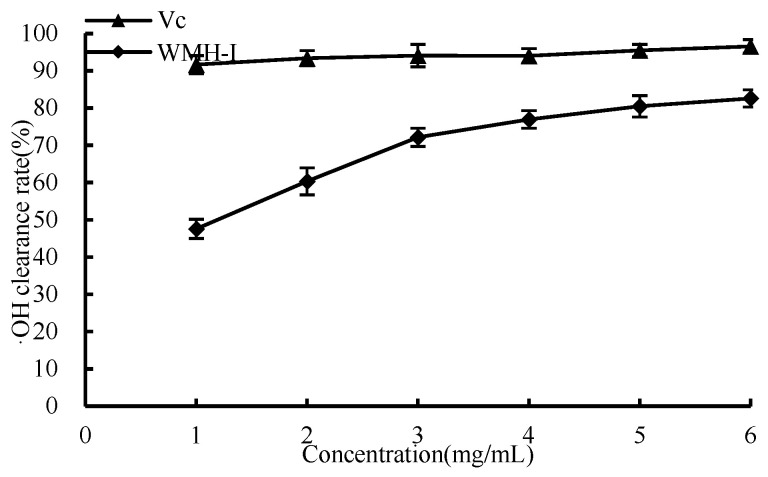
Clearance of ·OH by WMH-Ι and Vc.

**Figure 8 foods-13-01067-f008:**
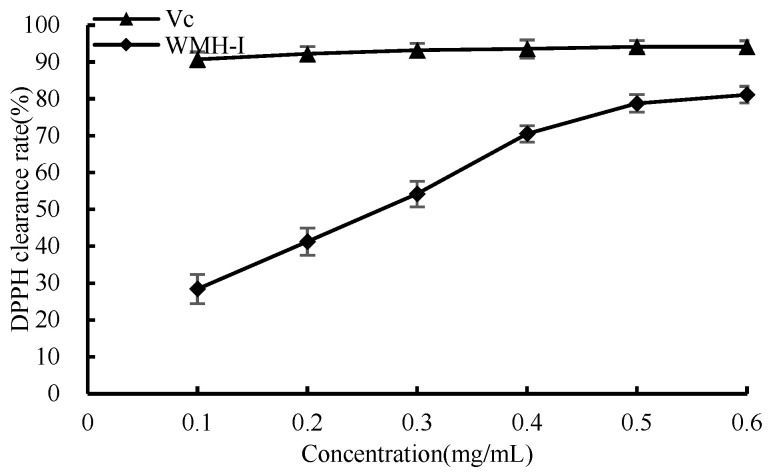
Scavenging rate of DPPH free radical by WMH-Ι and Vc.

**Figure 9 foods-13-01067-f009:**
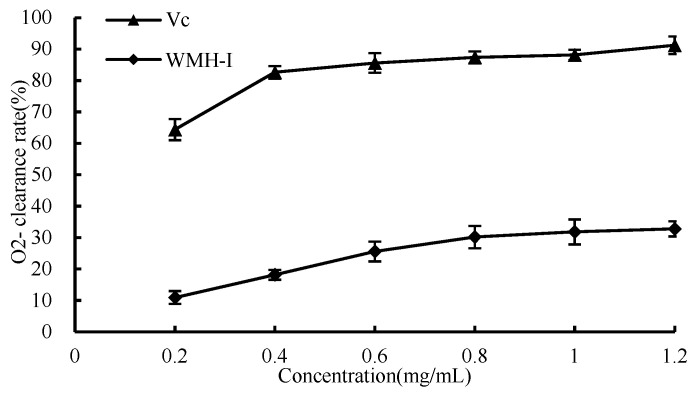
Clearance of superoxide anion by WMH-Ι and Vc.

**Figure 10 foods-13-01067-f010:**
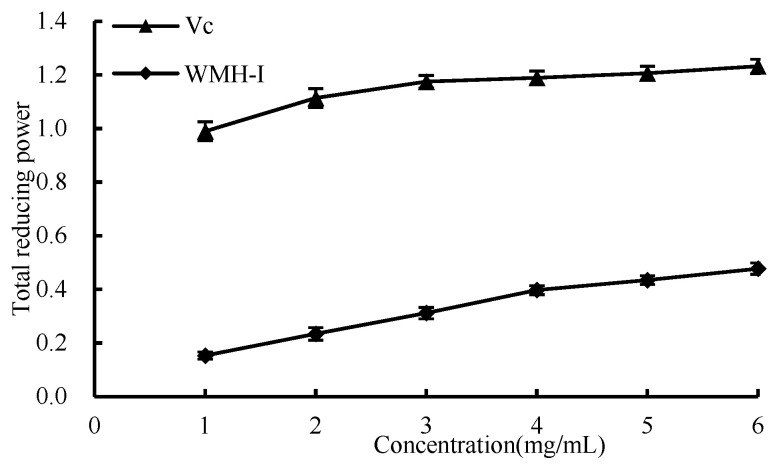
Total reducing power of WMH-Ι and Vc.

**Figure 11 foods-13-01067-f011:**
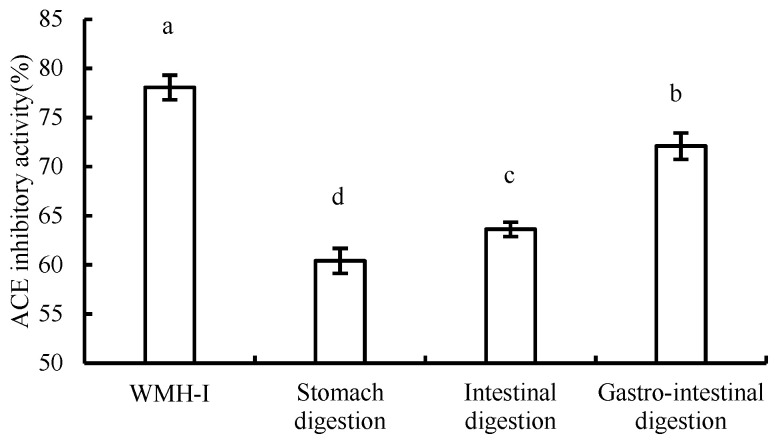
Comparison of ACE inhibitory activity during digestion. Data are expressed as means ± SD for three independent experiments; different letters marked significant differences as determined by one-way analysis of variation multiple test (*p* < 0.05).

**Figure 12 foods-13-01067-f012:**
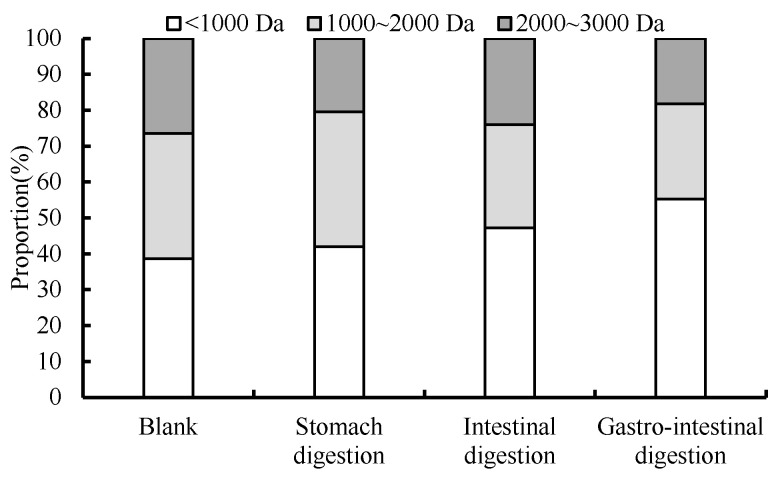
Molecular weight distribution of digestive products under different digestive processes.

**Table 1 foods-13-01067-t001:** Single-factor test factors and levels.

Influencing Factors	Level Setting				
	1	2	3	4	5
Amount of enzyme added (U/g)	2000	4000	6000	8000	10,000
Temperature (°C)	40	45	50	55	60
pH	8.5	9	9.5	10	10.5
Time (min)	60	90	120	150	180

**Table 2 foods-13-01067-t002:** Design factors and levels of Box–Behnken response surface test.

Influencing Factors		Level Setting		
		−1	0	1
A	pH	8.5	9	9.5
B	Temperature (°C)	50	55	60
C	Time (min)	60	120	150

**Table 3 foods-13-01067-t003:** In vitro detection methods of ACE inhibition rate.

Reagent	Amount Added (μL)		
	Experimental	Control	Blank
1 mol/L HCl	0	0	100
HHL	30	30	30
Peptide	10	0	10
Buffer	0	10	0
Operation 1	A water bath at 37 °C for 5 min		
ACE	10	10	10
Operation 2	A water bath at 37 °C for 30 min		
1 mol/L HCl	100	100	0

**Table 4 foods-13-01067-t004:** Box–Behnken response surface test design and results.

Test No.	A pH	B Temperature	C Time	Y ACE Inhibition Rate (%)
1	0	0	0	64.64
2	−1	0	1	60.03
3	0	0	0	63.34
4	1	0	−1	58.32
5	−1	1	0	57.06
6	−1	−1	0	60.26
7	0	−1	1	62.18
8	0	0	0	63.53
9	0	−1	−1	59.63
10	1	1	0	60.74
11	1	−1	0	60.48
12	−1	0	−1	57.83
13	0	0	0	63.66
14	0	0	0	63.16
15	1	0	1	62.96
16	0	1	1	61.37
17	0	1	−1	57.64

**Table 5 foods-13-01067-t005:** Regression analysis of the ACE inhibition rate response surface regression model.

Source	Sum of Square	df	Mean Square	F-Value	*p*-Value
Model	88.58	9	9.84	49.21	<0.0001 **
A-pH	6.70	1	6.70	33.49	0.0007 **
B-temperature	4.12	1	4.12	20.59	0.0027 **
C-time	21.52	1	21.52	107.59	<0.0001 **
AB	2.99	1	2.99	14.96	0.0061 **
AC	1.49	1	1.49	7.44	0.0294 *
BC	0.3481	1	0.3481	1.74	0.2286
A^2^	20.85	1	20.85	104.27	<0.0001 **
B^2^	13.73	1	13.73	68.63	<0.0001 **
C^2^	11.54	1	11.54	57.70	0.0001 **
Residual	1.40	7	0.20		
Lack of Fit	0.0704	3	0.0235	0.0707	0.9726
Pure Error	1.33	4	0.3324		
Cor Total	89.98	16			
R^2^ = 0.9844 Adj R^2^ = 0.9644 C.V.% = 0.73					

Note: *p* < 0.01 was considered extremely significant and indicated by **; *p* < 0.05 was considered significant and indicated by *.

**Table 6 foods-13-01067-t006:** Fractional ultrafiltration yield and ACE inhibition activities. Different letters represent significant differences (*p* < 0.05).

Group	Molecular Weight (Da)	Rate of Yield (%)	IC_50_ (mg/mL)
WMH	Stoste	100	0.401 ^b^
WMH-Ι	<3000	26.16	0.299 ^a^
WMH-ΙΙ	3000~10,000	39.82	0.652 ^c^
WMH-ΙΙΙ	>10,000	32.48	1.518 ^d^

## Data Availability

The original contributions presented in the study are included in the article, further inquiries can be directed to the corresponding authors.

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
