# Peer review of "Optimization of Extraction Process and Activity of Angiotensin-Converting Enzyme (ACE) Inhibitory Peptide from Walnut Meal"

_foods, 2024, doi:10.3390/foods13071067_

Round 1
Reviewer 1 Report
Comments and Suggestions for Authors
The manuscript considers a valuable contribution to utilizing walnut meal, a by-product of oil extraction, by focusing on extracting peptides with ACE inhibitory activity. The optimization of the extraction process using alkali-soluble acid precipitation and alkaline protease methods, alongside the evaluation of the antioxidant activities and stability under simulated gastrointestinal digestion, provides a comprehensive approach towards valorizing walnut meal for potential health benefits. The manuscript is a well-structured and insightful study that significantly contributes to food science and technology. However, some points must be clarified before the article can be accepted.
1. In the provided manuscript, the authors focus on optimizing peptide extraction processes, evaluating their ACE inhibitory and antioxidant activities, and assessing their stability under simulated gastrointestinal digestion. However, no information about the amino acid sequence of the isolated from walnut meal ACE inhibitory peptides is available. Identifying the amino acid sequences of bioactive peptides is crucial for understanding their mechanism of action and potential commercial synthesis, suggesting an area for future research based on this study.
2. In the manuscript, the authors provided the IC50 values for ACE inhibition as 1.156 mg/mL, 0.25 mg/mL, and 3.026 mg/mL for different types of antioxidant activities. However, these values refer to the capacity to remove free radicals (·OH, DPPH, ·O2-), not directly to ACE inhibition. In the context of ACE inhibition, the authors did not directly provide IC50 values for this specific activity in the available excerpts. Typically, the IC50 value for ACE inhibition would be a key indicator for evaluating the effectiveness of peptides as ACE inhibitors.
3. A comparison with existing ACE inhibitors, both synthetic and natural, could provide insight into the competitive advantages or complementary roles of walnut-derived peptides.
4. Please add and compered the obtained results with the reference: Liu, M., Du, M., Zhang, Y., Xu, W., Wang, C., Wang, K., & Zhang, L. (2013). Purification and identification of an ACE inhibitory peptide from walnut protein.. Journal of agricultural and food chemistry, 61 17, 4097-100 . https://doi.org/10.1021/jf4001378.
Author Response
Dear Reviewer:
Thank you very much for having our manuscript entitled “Optimization of Extraction Process and Activity of Angiotensin Converting Enzyme (ACE) Inhibitory Peptide from Walnut Meal” (manuscript number: Foods-2926902) reviewed in a timely and professional manner and for giving us an opportunity to revise it. We thank your warm work earnestly and deeply appreciate for yourcritical review of the manuscript with thoughtful and constructive comments. Accordingly, we have substantially revised our manuscript. Enclosed please find the revised version and point-by-point responses to reviewers’ comments are listed below.
This manuscript has not been published or submitted to another journal. All authors have seen the manuscript and approved to submit to your journal. The authors do not have any conflict of interest.
We sincerely hope that, with these extensive modifications and improvements, the format of manuscript is suitable for publication in “Foods”.
Once again, thank you very much for all your help and looking forward to hearing from you.
With best regards,
Sincerely yours,
Hui-qing Sun
State Key Laboratory of Food Nutrition and Safety, College of Food Science and Engineering, Tianjin University of Science and Technology, Tianjin, 300457, China
E-mail address: sunhuiqing@tust.edu.cn
The responses for comments are as followed and the revised manuscript is in the attachment.
The manuscript considers a valuable contribution to utilizing walnut meal, a by-product of oil extraction, by focusing on extracting peptides with ACE inhibitory activity. The optimization of the extraction process using alkali-soluble acid precipitation and alkaline protease methods, alongside the evaluation of the antioxidant activities and stability under simulated gastrointestinal digestion, provides a comprehensive approach towards valorizing walnut meal for potential health benefits. The manuscript is a well-structured and insightful study that significantly contributes to food science and technology. However, some points must be clarified before the article can be accepted.
Q1: In the provided manuscript, the authors focus on optimizing peptide extraction processes, evaluating their ACE inhibitory and antioxidant activities, and assessing their stability under simulated gastrointestinal digestion. However, no information about the amino acid sequence of the isolated from walnut meal ACE inhibitory peptides is available. Identifying the amino acid sequences of bioactive peptides is crucial for understanding their mechanism of action and potential commercial synthesis, suggesting an area for future research based on this study.
Response: Thank you for your professional and constructive comments. The core objective of this study is to optimize the enzymatic hydrolysis conditions and preliminarily explore the in vitro activity of the said substance, and on this basis to explore the activity changes of its enzymatic hydrolysis products and the optimization of related reaction conditions. However, as for the deeper details and mechanism analysis, we plan to leave them for further exploration in the future research work.
Q2: In the manuscript, the authors provided the IC50 values for ACE inhibition as 1.156 mg/mL, 0.25 mg/mL, and 3.026 mg/mL for different types of antioxidant activities. However, these values refer to the capacity to remove free radicals (·OH, DPPH, ·O2-), not directly to ACE inhibition. In the context of ACE inhibition, the authors did not directly provide IC50 values for this specific activity in the available excerpts. Typically, the IC50 value for ACE inhibition would be a key indicator for evaluating the effectiveness of peptides as ACE inhibitors.
Response: Thank you for your professional and constructive comments. I have updated the IC50 data for inhibitory ACE in Table 6.
Q3: A comparison with existing ACE inhibitors, both synthetic and natural, could provide insight into the competitive advantages or complementary roles of walnut-derived peptides.
Response: Thank you for your professional and constructive comments. The comparison with existing ACE inhibitor synthetic drugs is briefly summarized in the introduction. Ace-inhibiting peptides extracted from plants are safer and have no side effects for traditional blood pressure lowering drugs.
Q4: Please add and compered the obtained results with the reference: Liu, M., Du, M., Zhang, Y., Xu, W., Wang, C., Wang, K., & Zhang, L. (2013). Purification and identification of an ACE inhibitory peptide from walnut protein.. Journal of agricultural and food chemistry, 61 17, 4097-100 . https://doi.org/10.1021/jf4001378.
Response: Thank you for your professional and constructive comments. I have included references and comparisons in part of “3.3 Ultrafiltration Separation”.

Reviewer 2 Report
Comments and Suggestions for Authors
Optimization of Extraction Process and Activity of ACE2 Inhibitory Peptide from Walnut
The abstract provides a concise summary of the study's objectives, methods, results, and conclusions. It effectively highlights the significance of the research question and the key findings. However, the clarity of some sentences could be improved for better readability.
The introduction sets up the research question effectively and provides a clear rationale for the study. The relevance of the topic is well-established. However, there are minor issues with the use of terminology, such as italicizing "in vitro" and the need to introduce the complete term "angiotensin-converting enzyme (ACE)" upon first use.
The methods section is well-written and provides a detailed description of the experimental procedures. However, there are inconsistencies in the tense used throughout the section, and units of centrifugation force should be provided. Additionally, there are formatting issues with punctuation and spacing in certain areas that need to be addressed.
The results are presented clearly, but the figures lack self-explanatory labels, and there are inconsistencies in the capitalization of axis labels across different figures. Figure 11's presentation of significant differences in letters appears skewed and needs clarification.
The discussion section provides a thorough analysis of the results, but there are areas that require further clarification. For example, the rationale behind studying the antioxidant properties alongside ACE inhibition could be elaborated upon, including biochemical mechanisms linking hypertension and oxidative stress. Additionally, the writing style shifts noticeably after line 442, which may need revision for consistency.
Specific issues:
In vitro in italics.
Introduce the complete term the first time when ACE is used.
Line 60: revise the use of the word “water”. Hard to read sentence.
Line 62: a missing space after “;”
Materials and methods are written both in present tense and past.
Use g force units in the centrifugation methodology.
Line 122: period in other format: “…23)。In…”
Lines 174-179: homogenize the space between “°” and C in °C.
Line 191: stablish a concise sup topic title without action verbs.
Line 208: it is not clear which enzymes are the ones in the enzyme solution submitted to chromatographic analysis.
Line 215: incomplete sentence
Lines 222-223: hard to read.
Lines 262-263: repeated sentence.
Lines 292-293: use the superscript in R2.
Line 357: space missing in “…diseases(Obeng…”.
Line 387: change for a more formal word than “destroyed.”
Line 388: hard to read sentence.
Line 397: homogenize the form to mention figures, it is “Figure 12” in this line, but previously formatting would be Fig.12.
Line 410: in vivo in italics.
Line 418: space missing in “…activity(Garzón…”.
Line 443: space missing in “…blood(Hu…”.
Line 447: space missing in “…cells(Li…”.
Materials and methods:
· Homogenize the radical notion for ·OH vs ··OH and ··O2 vs O2−
· Homogenize the terms used in literature for results of antioxidative activity/capacity. Pleas identify which term is equivalent to the term used in the manuscript “DPPH rate of clearance” and use accordingly.
· The clearance rate equation is the same for all the antioxidative activity/capacity evaluations, I suggest mention the calculations once for all the methods.
Results:
· The figures are not self-explanatory. Please also homogenize the capital letter and the beginning of the words in the axis of all the figures
· Figure 11 presents the significant differences letters skewed.
Discussion
· Please explain more about: “In addition, the 420 ultrafiltration process helps to keep the peptide in its C-terminal position and promotes 421 inhibition (Silvestre, Silva, Silva, de Souza, Lopes, & Afonso, 2012)”.
· Please explain the rationale to study the antioxidant properties of the peptide fractions in this research that is focus on ACE inhibition, add information about biochemical mechanisms in which hypertension and ROS or prooxidative processes are linked.
· From line 442 the writing style (letters size and spacing) is different.
Comments on the Quality of English LanguageEnglish is not my strong point, but in this case it is difficult for me to understand some paragraphs. My recommendation is a better quality English manuscript.
Author Response
Dear Reviewer:
Thank you very much for having our manuscript entitled “Optimization of Extraction Process and Activity of Angiotensin Converting Enzyme (ACE) Inhibitory Peptide from Walnut Meal” (manuscript number: Foods-2926902) reviewed in a timely and professional manner and for giving us an opportunity to revise it. We thank the editors’ warm work earnestly and deeply appreciate the reviewers for their critical review of the manuscript with thoughtful and constructive comments. Accordingly, we have substantially revised our manuscript. Enclosed please find the revised version and point-by-point responses to reviewers’ comments are listed below.
This manuscript has not been published or submitted to another journal. All authors have seen the manuscript and approved to submit to your journal. The authors do not have any conflict of interest.
We sincerely hope that, with these extensive modifications and improvements, the format of manuscript is suitable for publication in “Foods”.
Once again, thank you very much for all your help and looking forward to hearing from you.
With best regards,
Sincerely yours,
Hui-qing Sun
State Key Laboratory of Food Nutrition and Safety, College of Food Science and Engineering, Tianjin University of Science and Technology, Tianjin, 300457, China
E-mail address: sunhuiqing@tust.edu.cn
The responses for comments are as followed. And the revised manuscript is in attachment.
Q1: The abstract provides a concise summary of the study's objectives, methods, results, and conclusions. It effectively highlights the significance of the research question and the key findings. However, the clarity of some sentences could be improved for better readability.
Response: Thank you for your professional and constructive comments. I have revised and polished the sentence with red mark in revised manuscript.
Q2: The introduction sets up the research question effectively and provides a clear rationale for the study. The relevance of the topic is well-established. However, there are minor issues with the use of terminology, such as italicizing "in vitro" and the need to introduce the complete term "angiotensin-converting enzyme (ACE)" upon first use.
Response: Thank you for your professional and constructive comments. We have revised and highlighted all "in vitro" references in the article and introduced the full term "angiotensin-converting enzyme" in the title and abstract.
Q3: The methods section is well-written and provides a detailed description of the experimental procedures. However, there are inconsistencies in the tense used throughout the section, and units of centrifugation force should be provided. Additionally, there are formatting issues with punctuation and spacing in certain areas that need to be addressed.
Response: Thank you for your professional and constructive comments. The unit of centrifugal force has been unified into g, and the format problem has also been modified.
Q4: The results are presented clearly, but the figures lack self-explanatory labels, and there are inconsistencies in the capitalization of axis labels across different figures. Fig.11's presentation of significant differences in letters appears skewed and needs clarification. Additionally, there are formatting issues with punctuation and spacing in certain areas that need to be addressed.
Response: Thank you for your professional and constructive comments. I have capitalized the beginning of the chart axis label and changed Figure 11.
Q5: The discussion section provides a thorough analysis of the results, but there are areas that require further clarification. For example, the rationale behind studying the antioxidant properties alongside ACE inhibition could be elaborated upon, including biochemical mechanisms linking hypertension and oxidative stress. Additionally, the writing style shifts noticeably after line 442, which may need revision for consistency.
Response: Thank you for your professional and constructive comments. Some biochemical mechanisms between hypertension and oxidative stress are added to the discussion section, and the writing style is unified after 442 lines.
Q6: Specific issues
Response: Thank you for your professional and constructive comments. We have modified the format and specific questions, and the modified parts have been marked in red in the manuscript.
Q7: Homogenize the radical notion for ·OH vs ··OH and ··O2 vs O2−
Response: Thank you for your professional and constructive comments. We think "·OH" is usually used to denote the hydroxyl radical, which is a free radical with unpaired electrons formed by covalent bonding of a single oxygen atom with a hydrogen atom. "··OH" may be a clerical error or a repetitive marker in an attempt to emphasize the properties of free radicals. While "··O2" may refer to peroxy free radicals, the correct expression should be "·O2·", that is, a radical formed by a single electron between two oxygen atoms in an oxygen molecule, "O2−" clearly indicates a superoxide anion, which is an ion formed after the acquisition of an electron on the oxygen molecule, rather than a free radical. We have changed all superoxide anions in the article to "O2−".
Q8: Homogenize the terms used in literature for results of antioxidative activity/capacity. Pleas identify which term is equivalent to the term used in the manuscript “DPPH rate of clearance” and use accordingly.
Response: Thank you for your professional and constructive comments. I have changed it to "DPPH rate of clearance".
Q9: The clearance rate equation is the same for all the antioxidative activity/capacity evaluations, I suggest mention the calculations once for all the methods.
Response: Thank you for your professional and constructive comments. We have removed the duplicate formulas and modified the continuity of the method.
Q10: The figures are not self-explanatory. Please also homogenize the capital letter and the beginning of the words in the axis of all the figures. Figure 11 presents the significant differences letters skewed.
Response: Thank you for your professional and constructive comments. We have capitalized the beginning of the chart axis label and changed Figure 11.
Q11: Please explain more about:“Inaddition, the 420 ultrafiltration process helps to keep the peptide in its C-terminal position and promotes 421 inhibition (Silvestre, Silva, Silva, de Souza, Lopes, & Afonso, 2012)”.
Response: Thank you for your professional and constructive comments. The authors suggest that the presence of hydrophobic amino acids such as tryptophan, tyrosine, phenylalanine and proline in the C-terminal portion of peptides with ACE inhibitory activity is necessary. The ultrafiltration process may result in the retention of peptides with structures that favor ACE inhibitory activity (aromatic amino acids or prolines). Therefore the sentence should be "In addition, the ultrafiltration process may also help to retain hydrophobic amino acids at the C-terminal position of the peptide and promote inhibition." We have modified it.
Q12: Please explain the rationale to study the antioxidant properties of the peptide fractions in this research that is focus on ACE inhibition, add information about biochemical mechanisms in which hypertension and ROS or prooxidative processes are linked.
Response: Thank you for your professional and constructive comments. Some amino acid residues that may exist in the structure of polypeptides (such as cysteine, tryptophan, tyrosine, etc.) can react with free radicals to prevent or delay the occurrence of oxidative chain reactions. By clearing the excess ROS produced in the body, oxidative stress can reduce the damage to cell membrane lipids, proteins and nucleic acids. ACE is a key enzyme in the renin-angiotensin system, which catalyzes the conversion of angiotensin I to angiotensin II, which has a strong vasoconstricting effect. Inhibition of ACE activity can effectively reduce the level of angiotensin II in plasma, thus playing a role in vasodilating, reducing cardiac load and reducing hypertension. Bioactive peptides bind and inhibit ACE activity through specific sequences, indirectly regulate blood pressure, and have potential preventive and therapeutic effects on hypertensive patients.
Q13: From line 442 the writing style (letters size and spacing) is different.
Response: Thank you for your professional and constructive comments. We have taken care of the whitespace, including the letter size and spacing.
Q14: Comments on the Quality of English Language. English is not my strong point, but in this case it is difficult for me to understand some paragraphs. My recommendation is a better quality English manuscript.
Response: Thank you for your professional and constructive comments. The manuscript had been revised by experts for English language editing to improve the quality and level.
